# QAConv: Question Answering on Informative Conversations

**Chien-Sheng Wu**[1], **Andrea Madotto**[2], **Wenhao Liu**[1], **Pascale Fung**[2], **Caiming Xiong**[1]

[1]Salesforce AI Research

[2]The Hong Kong University of Science and Technology

{wu.jason, wenhao.liu, cxiong}@salesforce.com

amadotto@connect.ust.hk, pascale@ece.ust.hk

## Abstract

This paper introduces QAConv[1], a new question answering (QA) dataset that uses conversations as a knowledge source. We focus on informative conversations, including business emails, panel discussions, and work channels. Unlike open-domain and task-oriented dialogues, these conversations are usually long, complex, asynchronous, and involve strong domain knowledge. In total, we collect 34,204 QA pairs, including multi-span and unanswerable questions, from 10,259 selected conversations with both human-written and machine-generated questions. We segment long conversations into chunks and use a question generator and a dialogue summarizer as auxiliary tools to collect multi-hop questions. The dataset has two testing scenarios, chunk mode and full mode, depending on whether the grounded chunk is provided or retrieved from a large pool of conversations. Experimental results show that state-of-the-art pretrained QA systems have limited zero-shot ability and tend to predict our questions as unanswerable. Finetuning such systems on our corpus can significantly improve up to 23.6% and 13.6% in both chunk mode and full mode, respectively.

## 1 Introduction

Having conversations is one of the most common ways to share knowledge and exchange information. Recently, many communication tools and platforms are heavily used with the increasing volume of remote working, and how to effectively retrieve information and answer questions based on past conversations becomes more and more important. In this paper, we focus on conversations such as business emails (e.g., Gmail), panel discussions (e.g., Zoom), and work channels (e.g., Slack). Different from daily chit-chat [1] and task-oriented dialogues [2], these conversations are usually long, complex, asynchronous, multi-party, and involve strong domain-knowledge. We refer to them as informative conversations and an example is shown in Figure 1.

However, QA research mainly focuses on document understanding (e.g., Wikipedia) not dialogue understanding, and dialogues have significant differences with documents in terms of data format and wording style [3, 4]. Existing work related to QA and conversational AI focuses on conversational QA [5, 6] instead of QA on conversations. Specifically, conversational QA has sequential dialogue-like QA pairs that are grounded on a short document paragraph, but what we are more interested in is to have QA pairs grounded on conversations, treating past dialogues as a knowledge source. QA on conversation has several unique challenges: 1) information is distributed across multiple speakers and scattered among dialogue turns; 2) Harder coreference resolution problem of speakers and entities, and 3) missing supervision as no training data in such format is available. The most related work

---

[1]Data and code are available at https://github.com/salesforce/QAConv

Submitted to the 35th Conference on Neural Information Processing Systems (NeurIPS 2021) Track on Datasets and Benchmarks. Do not distribute.

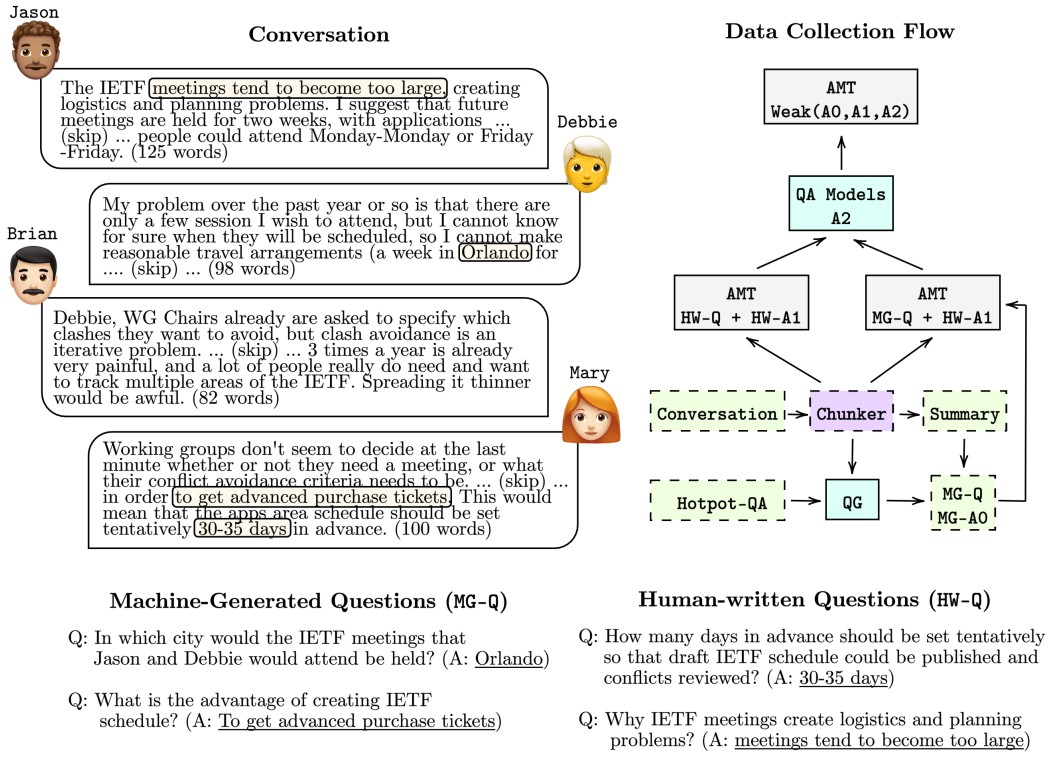

Figure 1: An example of question answering on conversations and the data collection flow.

to ours is the FriendsQA dataset [7] and the Molweni dataset [8]. The former is built on chit-chat transcripts of TV shows with only one thousand dialogues, and the latter is built on Ubuntu chat logs with short conversations. The dataset comparison to related work is shown in Table 1.

Therefore, we introduce QAConv dataset, sampling 10,259 conversations from email, panel, and channel data. The longest dialogue sample in our data has 19,917 words (or 32 speakers), coming from a long panel discussion. We segment long conversations into 18,728 shorter conversational chunks to collect human-written (HW) QA pairs or to modify machine-generated (MG) QA pairs from Amazon Mechanical Turk (AMT). We train a multi-hop question generator and a dialogue summarizer to obtain non-trivial QA pairs. We use QA models with predicted answers to identify uncertain samples and conduct an additional human verification stage. The data collection flow is shown in Figure 1. In total, we collect 34,204 QA pairs, including 5% unanswerable questions.

We construct two testing scenarios: 1) In the chunk mode, the corresponding conversational chunks are provided to answer questions, similar to the SQuAD dataset [9]; 2) In the full mode, a conversational-retrieval stage is required before answering questions, similar to the open-domain QA dataset [10]. We explore several state-of-the-art QA models such as the span extraction RoBERTa-Large model [11] trained on SQuAD 2.0 dataset, and the generative UnifiedQA model [12] trained on 20 different QA datasets. We investigate the statistic-based BM25 [13] retriever and the neural-based dense passage retriever [14] trained on Wikipedia (DPR-wiki) as our base conversational retrievers. We show zero-shot and finetuning performances in both modes and conduct improvement study and error analysis.

The main contributions of our paper are threefold: 1) QAConv provides a new testbed for QA on informative conversations including emails, panel discussions, and work channels; 2) We are the first to incorporate multi-hop question generation (QG) model into QA data collection, and we show the effectiveness of such approach in human evaluation; 3) We show the potential of treating long conversations as a knowledge source and point out a performance gap in existing QA models trained with documents versus our proposed QAConv.

Table 1: Dataset comparison with existing datasets.

| | QAConv | | **Molweni** | **DREAM** | **FriendsQA** |
|---|---|---|---|---|---|
| | Full | Chunk | | | |
| Source | Email, Panel, Channel | | Channel | Chit-chat | Chit-chat |
| Domain | General | | Ubuntu | Daily | TV show |
| Formulation | Multi-span/Unanswerable | | Span/Unanswerable | Multiple choice | Span |
| Questions | 34,204 | | 30,066 | 10,197 | 10,610 |
| Dialogues | 10,259 | 18,728 | 9,754 | 6,444 | 1,222 |
| Avg/Max Words | 568.8 / 19,917 | 303.5 / 6,787 | 104.4 / 208 | 75.5 / 1,221 | 277.0 / 2,438 |
| Avg/Max Speakers | 2.8 / 32 | 2.9 / 14 | 3.5 / 9 | 2.0 / 2 | 3.9 / 15 |

## 2  QAConv **Dataset**

Our dataset is collected in four stages: 1) selecting and segmenting informative conversations, 2) generating question candidates by multi-hop QG models, 3) crowdsourcing question-answer pairs on those conversations/questions, and 4) conducting quality verification and data splits.

### 2.1  Data Collection

#### 2.1.1  Selection and Segmentation

First, we use the British Columbia conversation corpora (BC3) [15] and the Enron Corpus [16] to represent business email use cases. The BC3 is a subset of the World Wide Web Consortium's (W3C) sites that are less technical. We sample threaded Enron emails from [17], which were collected from the Enron Corporation. Second, we select the Court corpus [18] and the Media dataset [19] as panel discussion data. The Court data is the transcripts of oral arguments before the United States Supreme Court. The Media data is the interview transcriptions from National Public Radio and Cable News Network. Third, we choose the Slack chats [20] to represent work channel conversations. The Slack data was crawled from several public software-related development channels such as *pythondev#help* Full data statistics of each source are shown in Table 2. All data we use is publicly available and their license and privacy information are shown in the Appendix.

One of the main challenges in our dataset collection is the length of input conversations and thus resulting in very inefficient for crowd workers to work on. For example, on average there are 13,143 words per dialogues in the Court dataset, and there is no clear boundary annotation in a long conversation of a Slack channel. Therefore, we segment long dialogues into short chunks by a turn-based buffer to assure that the maximum number of tokens in each chunk is lower than a fixed threshold, i.e., 512. For the Slack channels, we use the disentanglement script from [20] to split channel messages into separated conversational threads, then we either segment long threads or combine short threads to obtain the final conversational chunks.

#### 2.1.2  Multi-hop Question Generation

To get more non-trivial questions that require reasoning (i.e., answers are related to multiple sentences or turns), we leverage a question generator and a dialogue summarizer to generate multi-hop questions. We have two hypotheses: 1) QG models trained on multi-hop QA datasets can produce multi-hop questions, and 2) QG models taking dialogue summary as input can generate high-level questions. By the first assumption, we train a T5-Base [21] model on HotpotQA [22], which is a QA dataset featuring natural and multi-hop questions, to generate questions for our conversational chunks. By the second hypothesis, we first train a BART [23] summarizer on News [24] and dialogue summarization corpora [25] and run QG models on top of the generated summaries.

We filter out generated questions that 1) a pretrained QA model can have consistent answers, and 2) a QA model has similar answers grounded with conversations or summaries. Note that our QG model has "known" answers since it is trained to generate questions by giving a text context and an extracted entity. We hypothesize that these questions are trivial questions in which answers can be easily found, and thus not of interesting for our dataset.

Table 2: Dataset statistics of different dialogue sources.

| | BC3 | | Enron | | Court | |
|---|---|---|---|---|---|---|
| | Full | Chunk | Full | Chunk | Full | Chunk |
| Questions | 164 | | 8096 | | 9456 | |
| Dialogues | 40 | 84 | 3,257 | 4,220 | 125 | 4,923 |
| Avg/Max Words | 514.9 / 1,236 | 245.2 / 593 | 383.6 / 69,13 | 285.8 / 6,787 | 13,143.4 / 19,917 | 330.7 / 1,551 |
| Avg/Max Speakers | 4.8 / 8 | 2.7 / 6 | 2.7 / 10 | 2.2 / 8 | 10.3 / 14 | 2.7 / 7 |

| | Media | | Slack | |
|---|---|---|---|---|
| | Full | Chunk | Full | Chunk |
| Questions | 9,155 | | 5,599 | |
| Dialogues | 699 | 4,812 | 6,138 | 4,689 |
| Avg/Max Words | 2,009.6 / 11,851 | 288.7 / 537 | 247.2 / 4,777 | 307.2 / 694 |
| Avg/Max Speakers | 4.4/ 32 | 2.4 / 11 | 2.5 / 15 | 4.3 / 14 |

### 2.1.3 Crowdsourcing QA Pairs

We use two strategies to collect QA pairs, human writer and machine generator. We first ask crowd workers to read partial conversations, and then we randomly assign two settings: 1) writing QA pairs themselves or 2) selecting one recommended machine-generated question to answer. We apply several on-the-fly constraints to control the quality of the collected QA pairs: 1) questions should have more than 6 words with a question mark in the end, and at least 10% words have to appear in source conversations; 2) questions and answers cannot contain first-person and second-person pronouns (e.g., I, you, etc.); 3) answers have to be less than 20 words and all words have to appear in source conversations, but not necessarily from the same text span.

We randomly select four MG questions from our question pool and ask crowd workers to answer one of them, without providing our predicted answers. They are allowed to modify questions if necessary. To collect unanswerable questions, we ask crowd workers to write questions with at least three entities mentioned in the given conversations but they are not answerable. We pay crowd workers roughly $8-10 per hour, and the average time to read and write one QA pair is approximately 4 minutes.

### 2.1.4 Quality Verification and Data Splits

We design a filter mechanism based on different potential answers: human writer's answers, answer from existing QA models, and QG answers. If all the answers have pairwise fuzzy matching ratio (FZ-R) scores [2] lower than 75%, we then run another crowdsourcing round and ask crowd workers to select one of the following options: A) the QA pair looks good, B) the question is not answerable, C) the question has a wrong answer, and D) the question has a right answer but I prefer another answer. We run this step on around 40% samples which are uncertain. We filter the questions of the (C) option and add answers of the (D) option into ground truth. In questions marked with option (B), we combine them with the unanswerable questions that we have collected. In addition, we include 1% random questions (questions that are sampled from other conversations) to the same batch of data collection as qualification test. We filter crowd workers' results if they fail to indicate such question as an option (B). Finally, we split the data into 80% training, 10% validation, and 10% testing by sampling within each dialogue source, resulting in 27,287 training samples, 3,414 validation samples, and 3503 testing samples. There are 4.7%, 4.8%, 5.8% unanswerable questions in train, validation, and test split, respectively.

## 2.2 QA Analysis

In this section, we analyze our collected questions and answers. We first investigate question type distribution and we compare human-written questions and machine-generated questions. We then analyze answers by an existing named-entity recognition (NER) model and a constituent parser.

### 2.2.1 Question Analysis

**Question Type.** We show the question type tree map in Figure 2 and the detailed comparison with other datasets in the Appendix (Table 10). In QAConv, the top 5 question types are what-question (29%), which-question (27%), how-question (12%), who-question (10%), and when-question

---

[2] https://pypi.org/project/fuzzywuzzy

**What** — What order must the list be sorted? What contract do Dylan need? What region of California is the Van from? What way the Hiroko was add the media? What became a post WWI food staple? What Ted Kaptchuk said about placebo?

**What does** — What does Johnson say is "fairly complex"? What does Loris want to output JSON as?

**What did** — What did the judge impose a tax on that day?

**How** — How would Demetrice make a copy of the list? how LSP is supposed to work with langs? How Cherrie tried to move ? How wide is the vent in the volcano?

**Who** — Who's the last person to be back to address the issue with Akzo? Who warded the message to Kevin?

**What is** — What is proposed to be the goal? What does 'f' stand for in apply-f? What is the name of the petitioner in the case? What does Joan want to do while in Sunriver? What is the name of the Chief Justice?

**What was** — What was the Name of Cybersecurity Professor at the GIoT? What was the Warwick?

**What type** — What type of material will Bill have an allergic reaction?

**How many** — How many planets are there? How many tickets Eris mentioned to Paul?

**Who is** — Who is the litigation manager mentioned by carol?

**When** — When Dylan is going back to Dome? When William wrote the first paper? When Mark spoke with Cynthia?

**When did** — When did congress enact 2242?

**Which** — Which age groups are drug dealers? Which girl is learning HtDP? Which other person is Ida discussing? Which game was mentioned in the passage? which simple code is worked by Sheri at first? Which item does Vince ask Shirley to order?

**Which person** — Which person is talking to the Chief Justice? Which person is dating a guy from CU?

**Which year** — Which year does John reference regarding the Utility M&A?

**Where** — Where does Rob Robert L. Bradley Jr. work? Where is Neal Conan from? Where was the luggage placed?

**Why** — Why does the piece of code feel inefficient? Why will the speaker send the drafts to Kay?

**Which is** — Which is a fundamental read according to Terrence? Which is written by Zimin Lu? which is the background expander found said by Odis?

**Which type** — Which type of authentication is Dawn using?

**Which case** — Which city does Taniesha Woods work from ?

**Other** — In which industry lynda need the survey about developers? Jason wrote stories for which paper? Transactions will be between which two entities?

Figure 2: Question type tree map and examples (Best view in color).

Table 3: HW questions v.s. MG questions: Ratio and human evaluation.

| Source | Question Generator | | | | Human Writer | |
|---|---|---|---|---|---|---|
| Questions | 14,076 (41.2%) | | | | 20,128 (58.8%) | |
| Type | 100 | 81-99 | 51-79 | 0-50 | Ans. | Unans. |
| Ratio | 33.56% | 19.92% | 24.72% | 21.80% | 91.39% | 8.61% |
| Avg. Words | **12.94** ($\pm$5.14) | | | | 10.98 ($\pm$3.58) | |
| Fluency | **1.808** | | | | 1.658 | |
| Complexity | **0.899** | | | | 0.674 | |
| Confidence | 0.830 | | | | **0.902** | |

(6%). Comparing to SQuAD 2.0 (49% what-question), our dataset have a more balanced question distribution. The question distribution of unanswerable questions is different from the overall distribution. The top 5 unanswerable question types are what-question (45%), why-question (15%), how-question (12%), which-question (10%), and when-question (8%), where the why-question increases from 3% to 15%.

**Human Writer v.s. Machine Generator.** As shown in Table 3, there are 41.2% questions are machine-generated questions. Since we still give crowd workers the freedom to modify questions if necessary, we cannot guarantee these questions are unchanged. We find that 33.56% of our recommended questions have not been changed (100% fuzzy matching score) and 19.92% of them are slightly modified (81%-99% fuzzy matching score). To dive into the characteristics and differences of these two question sources, we further conduct the human evaluation by sampling 200 conversation chunks randomly. We select chunks that have QG questions unchanged (i.e., sampling from the 33.56% QG questions). We ask three annotators to first write an answer to the given question and conversation, then label fluency (how fluent and grammatically correct the question is, from 0 to 2), complexity (how hard to find an answer, from 0 to 2), and confidence (whether they are confident with their answer, 0 or 1). More details of each evaluation dimension are shown in the Appendix. The results in Table 3 indicate that QG questions are longer, more fluent, more complex, and crowd workers are less confident that they are providing the right answers. This observation further confirmed our hypothesis that the multi-hop question generation strategy is effective to collect harder QA examples.

### 2.2.2 Answer Analysis

Following [9], we used Part-Of-Speech (POS) [26] and Spacy NER taggers to study answers diversity. Firstly, we use the NER tagger to assign an entity type to the answers. However, since our answers are not necessary to be an entity, those answers without entity tags are then pass to the POS tagger, to extract the corresponding phrases tag. In Table 4, we can see that Noun phrases make up 30.4% of the

Table 4: Answer type analysis.

| Answer type | Percentage | Example |
|---|---|---|
| Prepositional Phrase | 1.3% | with 'syntax-local-lift-module' |
| Nationalities or religious | 1.3% | white Caucasian American |
| Monetary values | 1.6% | $250,000 |
| Clause | 5.4% | need to use an external store for state |
| Countries, cities, states | 8.9% | Chicago |
| Other Numeric | 9.6% | page 66, volume 4 |
| Dates | 9.6% | 2020 |
| Organizations | 11.4% | Drug Enforcement Authority |
| People, including fictional | 12.5% | Tommy Norment |
| Noun Phrase | 30.4% | the Pulitzer Prize |

data; followed by People, Organization, Dates, other numeric, and Countries; and the remaining are made up of clauses and other types. Full category distribution is shown in the Appendix (Figure 3).

## 2.3 Chunk Mode and Full Mode

The main difference between the two modes is whether the conversational chunk we used to collect QA pairs is provided or not. In the chunk mode, our task is more like a traditional machine reading comprehension task that answers can be found (or cannot be found) in a short paragraph, usually less than 500 words. In the full mode, on the other hand, we usually need an information retrieval stage before the QA stage. For example, in the Natural Question dataset [27], they split Wikipedia into millions of passages and retrieve the most relevant one to answer.

We define our full mode task with the following assumptions: 1) for the email and panel data, we assume to know which dialogue a question is corresponding to, that is, we only search chunks within the dialogue instead of all the possible conversations. This is simpler and more reasonable because each conversations are independent; 2) for slack data, we assume that we only know which channel a question is belongs to but not the corresponding thread, so the retrieval part has to be done in the whole channel. Even though a question could be ambiguous in the full mode due to the way of data collection, we find that most of our collected questions are self-contained and entity-specific. Also, for open-domain question answering task, it has been shown that the recall metric can be more important than the precision metric [28].

## 3 Experimental Results

### 3.1 State-of-the-art Baselines

There are two categories of question answering models: span-based extractive models which predict answers' start and end positions, and free-form text generation models which directly generate answers token by token. We evaluate all of them on both zero-shot and finetuned settings, and both chunk mode and full mode with retrievers. In addition, we run these models on the Molweni [8] dataset for comparison, and find out all our baselines outperform the DADgraph [29] model, the current best reported model using expensive discourse annotation on graph neural network. We show the Molweni results in the Appendix (Table 9).

### 3.1.1 Span-based Models

We use several pretrained language models finetuned on the SQuAD 2.0 dataset as span extractive baselines. We use uploaded models from huggingface [30] library. DistilBERT [31] is a knowledge-distilled version with 40% size reduction from the BERT model, and it is widely used in mobile devices. The BERT-Base and RoBERTa-Base [11] model are evaluated as the most commonly used in the research community. We also run the BERT-Large and RoBERTa-Large models as stronger baselines. We use the whole-word masking version of BERT-Large instead of the token masking one from the original paper since it performs better.

Table 5: Evaluation results: Chunk mode on the test set.

| | Zero-Shot | | | Finetune | | |
|---|---|---|---|---|---|---|
| | **EM** | **F1** | **FZ-R** | **EM** | **F1** | **FZ-R** |
| Human Performance | 79.99 | 89.87 | 92.33 | - | - | - |
| DistilBERT-Base (SQuAD 2.0) | 46.50 | 52.79 | 63.30 | 63.69 | 73.94 | 79.30 |
| BERT-Base (SQuAD 2.0) | 42.73 | 49.67 | 60.99 | 66.37 | 76.29 | 81.25 |
| BERT-Large (SQuAD 2.0) | **61.06** | **68.11** | **74.98** | 72.85 | 81.65 | 85.59 |
| RoBERTa-Base (SQuAD 2.0) | 57.75 | 64.53 | 72.40 | 71.14 | 80.36 | 84.52 |
| RoBERTa-Large (SQuAD 2.0) | 59.04 | 66.54 | 73.80 | **74.62** | **83.65** | **87.38** |
| T5-Base (UnifiedQA) | 57.75 | 69.90 | 76.31 | 71.20 | 80.92 | 84.74 |
| T5-Large (UnifiedQA) | 64.83 | 75.73 | 80.59 | 73.54 | 83.03 | 86.61 |
| T5-3B (UnifiedQA) | **66.77** | **76.98** | **81.77** | **75.21** | **84.14** | **87.47** |
| T5-11B (UnifiedQA) | 51.13 | 66.19 | 71.68 | - | - | - |
| GPT-3 | 53.72 | 67.45 | 72.94 | - | - | - |

Table 6: Answerable/Unanswerable results: Chunk Mode on the test set.

| | Zero-Shot | | | | Finetune | | | |
|---|---|---|---|---|---|---|---|---|
| | Ans. | | Unans. | Binary | Ans. | | Unans. | Binary |
| | **EM** | **F1** | **Recall** | **F1** | **EM** | **F1** | **Recall** | **F1** |
| Human Performance | 80.46 | 90.95 | 72.27 | 71.01 | - | - | - | - |
| DistilBERT-Base (SQuAD) | 44.47 | 51.15 | 79.70 | 22.41 | 65.01 | 75.89 | 42.08 | 42.59 |
| BERT-Base (SQuAD2) | 40.23 | 47.59 | 83.66 | 21.80 | 67.62 | 78.15 | 46.04 | 44.59 |
| BERT-Large (SQuAD2) | **59.98** | **67.64** | 78.71 | 30.26 | 74.19 | 83.52 | 50.99 | 53.66 |
| RoBERTa-Base (SQuAD2) | 56.44 | 63.64 | 79.21 | 27.56 | 72.71 | 82.49 | 45.54 | 47.78 |
| RoBERTa-Large (SQuAD2) | 57.16 | 65.13 | **89.60** | **30.89** | **76.01** | **85.59** | 51.98 | **55.64** |
| T5-Base (UnifiedQA) | 62.61 | 75.79 | 0.0 | 0.0 | 74.31 | 84.85 | 34.19 | 44.29 |
| T5-Large (UnifiedQA) | 70.29 | 82.11 | 0.0 | 0.0 | 76.75 | 87.04 | 35.29 | 47.17 |
| T5-3B (UnifiedQA) | **72.39** | **83.46** | 0.0 | 0.0 | **77.65** | **87.33** | 46.32 | **57.01** |

### 3.1.2 Free-form Models

We run several versions of UnifiedQA models [12] as strong generative QA baselines. UnifiedQA is based on T5 model [21], a language model that has been pretrained on 750GB C4 text corpus. UnifiedQA further finetuned T5 models on 20 existing QA corpora spanning four diverse formats, including extractive, abstractive, multiple-choice, and yes/no questions. It has achieved state-of-the-art results on 10 factoid and commonsense QA datasets. We finetune UnifiedQA on our datasets with T5-Base, T5-Large size, and T5-3B. We report T5-11B size for the zero-shot performance. We also test the performance of GPT3 [32], where we design the prompt as concatenating a training example from CoQA [5] and our test samples. The prompt we used is shown in the Appendix (Table 15).

### 3.1.3 Retrieval Models

Two retrieval baselines are investigated in this paper: BM25 and DPR-wiki. The BM25 retriever is a bag-of-words retrieval function weighted by term frequency and inverse document frequency. The DPR-wiki model is a BERT-based [33] dense retriever model trained for open-domain QA tasks, learning to retrieve the most relevant Wikipedia passage. We trained the DPR-wiki model by sharing the passage encoder and question encoder, and we reduce the dimension of the dense representations from 768 to 128 with one fully connected layer to speed up whole retrieval process.

### 3.2 Evaluation Metrics

We follow the standard evaluation metrics in the QA community: exact match (EM) and F1 scores. The EM score is a strict score that predicted answers have to be the same as the ground truth answers. The F1 score is calculated by tokens overlapping between predicted answers and ground truth answers. In addition, we also report the FZ-R scores, which used the Levenshtein distance to calculate the differences between sequences. We follow [9] to normalize the answers in several ways: remove stop-words, remove punctuation, and lowercase each character. We add one step with the *num2words* and *word2number* libraries to avoid prediction difference such as "2" and "two".

Table 7: Retriever results: BM25 and DPR on the test set.

| | R@1 | R@3 | R@5 | R@10 |
|---|---|---|---|---|
| BM25 | **0.584** | **0.755** | **0.801** | **0.852** |
| DPR-wiki | 0.432 | 0.596 | 0.661 | 0.751 |

Table 8: Evaluation results: Full mode with BM25 on the test set.

| BM25 | Zero-Shot | | | Finetune | | |
|---|---|---|---|---|---|---|
| | **EM** | **F1** | **FZ-R** | **EM** | **F1** | **FZ-R** |
| DistilBERT-Base (SQuAD 2.0) | 33.66 | 38.19 | 52.28 | 43.51 | 52.12 | 62.63 |
| BERT-Base (SQuAD 2.0) | 30.80 | 35.80 | 50.50 | 44.62 | 52.91 | 63.50 |
| BERT-Large (SQuAD 2.0) | **42.19** | **47.59** | **59.41** | 48.99 | 56.60 | 66.40 |
| RoBERTa-Base (SQuAD 2.0) | 41.11 | 46.15 | 58.35 | 48.42 | 56.24 | 66.08 |
| RoBERTa-Large (SQuAD 2.0) | 41.39 | 46.75 | 58.67 | **50.24** | **57.80** | **67.57** |
| T5-Base (UnifiedQA) | 39.68 | 49.76 | 60.51 | 48.56 | 56.38 | 66.01 |
| T5-Large (UnifiedQA) | 44.08 | 53.17 | 63.17 | 49.64 | 57.58 | 67.36 |
| T5-3B (UnifiedQA) | **45.87** | **55.24** | **64.83** | **51.44** | **58.80** | **68.10** |

## 3.3 Performance Analysis

### 3.3.1 Chunk Mode

We first estimate human performance by asking crowd workers to answer our QA pairs in the test set. We collect two answers for each question and select one that has a higher FZ-R score. As the chunk mode results shown in Table 5, UnifiedQA T5 model with 3B size achieves 66.77% zero-shot EM score and 75.21% finetuned EM score, which is close to human performance by less than 5%. This observation matches the recent trend that large-scale pretrained language model finetuned on aggregated datasets of a specific downstream task (e.g., QA tasks [12] and dialogue task [4]) can show state-of-the-art performance by knowledge transferring.

Those span-based models, meanwhile, achieve good performance with a smaller model size. BERT-Base model has the largest improvement gain by 23.64 EM score after finetuning. BERT-Large model with word-masking pretraining achieves 61.06% zero-shot EM score, and RoBERTa-Large model trained on SQuAD 2.0 achieves 74.62% if we further finetune it on our training set. We find that UnifiedQA T5 model with 11B parameters cannot achieve performance as good as the 3B model, one potential reason is that the model is too big and has not been well-trained by [12]. The GPT-3 model with CoQA prompt can at most achieve 53.72% zero-shot performance with our current prompt design.

We further check the results difference between answerable and unanswerable questions in Table 6. The UnifiedQA models outperform span-based models among the answerable questions, however, they are not able to answer any unanswerable questions and keep predicting some "answers". More interesting, we observe that those span-based models perform poorly on an answerable question, achieving high recall but low F1 on unanswerable questions for the binary setting (predict answerable or unanswerable). This suggests that existing span-based models tend to predict our task as unanswerable, revealing the weakness of their dialogue understanding abilities.

### 3.3.2 Full Mode

The retriever results are shown in Table 7, in which we find that BM25 outperforms DPR-wiki by a large margin in our dataset on the recall@$k$ measure, where we report $k = 1, 3, 5, 10$. The two possible reasons are that 1) the difference in data distribution between Wikipedia and conversation is large and DPR is not able to properly transfer to unseen documents, and 2) questions in QAConv are more specific to those mentioned entities, which makes the BM25 method more reliable. We show the full mode results in Table 8 using BM25 (DPR-wiki results in Table 11). We use only one retrieved conversational chunk as input to the trained QA models. As a result, the performance of UnifiedQA (T5-3B) drops around 20% in the zero-shot setting, and the finetuned results of RoBERTa-Large drop by 24.4% as well, suggesting a serious error propagation issue in the full mode.

## 4 Error Analysis

We first check what kinds of QA samples in the test set are improved the most while finetuning on our training data under the chunk mode. We check those samples which are not exactly matched in the RoBERTa-Large zero-shot experiment but become correct after finetuning. We find that 75% of such samples are incorrectly predicted to be unanswerable, which is consistent with the results in Table 6. Next, we analyze the error prediction after finetuning. We find that 35.5% are what-question errors, 18.2% are which-question errors, 12.1% are how-question errors, and 10.3% are who-question errors. We also sample 100 QA pairs from the errors which have an FZ-R score lower than 50% and manually check the predicted answers. We find out that 20% of such examples are somehow reasonable (e.g., UCLA v.s. University of California, Jay Sonneburg v.s. Jay), 31% are predicted wrong answers but with correct entity type (e.g., Eurasia v.s. China, Susan Flynn v.s. Sara Shackleton), 38% are wrong answers with different entity types (e.g., prison v.s. drug test, Thanksgiving v.s., fourth quarter), and 11% are classified as unanswerable questions wrongly.

## 5 Related Work

QA datasets can be categorized into four groups. The first one is cloze-style QA where a model has to fill in the blanks. For example, the Children's Book Test [34] and the Who-did-What dataset [35]. The second one is reading comprehension QA where a model picks the answers for multiple-choice questions or a yes/no question. For examples, RACE [36] and DREAM [37] datasets. The third one is span-based QA, such as SQuAD [9] and MS MARCO [38] dataset, where a model extracts a text span from the given context as the answer. The fourth one is open-domain QA, where the answers are selected and extracted from a large pool of passages, e.g., the WikiQA [39] and Natural Question [27] datasets.

Conversation-related QA tasks have focused on asking sequential questions and answers like a conversation and grounded on a short passage. CoQA [5] and QuAC [6] are the two most representative conversational QA datasets under this category. CoQA contains conversational QA pairs, free-form answers along with text spans as rationales, and text passages from seven domains. QuAC collected data by a teacher-student setting on Wikipedia sections and it could be open-ended, unanswerable, or context-specific questions. Closest to our work, Dream [37] is a multiple-choice dialogue-based reading comprehension examination dataset, but the conversations are in daily chit-chat domains between two people. FriendsQA [7] is compiled from transcripts of the TV show Friends, which is also chit-chat conversations among characters and only has around one thousand dialogues. Molweni [8] is built on top of Ubuntu corpus [40] for machine reading comprehension task, but its conversations are short and focused on one single domain, and their questions are less diverse due to their data collection strategy (10 annotators).

In general, our task is also related to conversations as a knowledge source. The dialogue state tracking task in task-oriented dialogue systems can be viewed as one specific branch of this goal as well, where tracking slots and values can be re-framed as a QA task [41, 42], e.g., "where is the location of the restaurant?". Moreover, extracting user attributes from open-domain conversations [43], getting to know the user through conversations, can be marked as one of the potential applications. The very recently proposed query-based meeting summarization dataset, QMSum [44], can be viewed as one application of treating conversations as database and conduct an abstractive question answering task.

## 6 Conclusion

QAConv is a new dataset that conducts QA on informative conversations such as emails, panels, and channels. It has 34,204 questions including span-based, free-form, and unanswerable questions. We show the unique challenges of our tasks in both chunk mode with oracle partial conversations and full mode with a retrieval stage. We find that state-of-the-art QA models have limited zero-shot performance and tend to predict our answerable QA pairs as unanswerable, and they can be improved significantly after finetuning. QAConv is a new testbed for QA on conversation tasks and conversations as a knowledge source research.

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
