# OpenReview forum: "QAConv: Question Answering on Informative Conversations"
_NeurIPS.cc/2021/Track/Datasets_and_Benchmarks/Round1 — Submitted to NeurIPS 2021 Datasets and Benchmarks Track (Round 1)_

### Official Review · Reviewer_AzsR · 2021-06-29
**OK but not that challenging conversational QA dataset**

**Rating:** 5
**Confidence:** 4
**Correctness:** The construction of the dataset is re…

**Strengths:**

1. Introducing the question generation and dialogue summarizer in the annotation process is effective and quite novel.
2. The documentation of the dataset is relatively thorough.

**Weaknesses:**

1. The proposed dataset is less challenging to the recent pre-trained language models, where the gap between the human performance and the best system is less than 5%.
2. The authors only report the results on the test set but not on the development set. Also, the zero-shot setting is confusing. The authors did not illustrate why such setting is necessary.
3. There are many points left unclear in the paper, which should be addressed during the rebuttal period. Please see 'major comments' at the end of this review. (I am open to discuss them.)
4. There are some typos and grammatical errors in the paper, which should be avoided. (This does not affect my rating on this paper.)

**Additional Feedback:**

Major comments:
1. The authors claim that their dataset includes multi-span in the abstract (line 6) and Table 1. However, in Figure 1, I did not see such examples. Could you explain why your dataset features 'multi-span'? If this is the case, a clear illustration on how to evaluate multi-span answer (maybe concatenate all spans?) should be included in Section 3.2.
2. line 89-92: Are there any particular reasons that you use T5-base for question generation but use BART as the dialogue summarizer?
3. It is quite confusing to only show the test set results. Why are the development set scores missing? For future work comparisons, it is necessary to release the development set scores first.
4. I am quite disappointed that current state-of-the-art models only have a minor gap to human performance. This was also indicated by the authors in line 224-226. Specifically, the best finetuned model achieves 75.2/84.1/87.4 on the test set, while human performance is 79.9/89.8/92.3. I believe that after several task-specific modifications and hyper-parameter tuning, the gap between them can be narrowed further. These results suggest that the proposed dataset is not that challenging.
5. The zero-shot experimental setting is confusing. As the training data is already available in QAConv, the authors should explain why the zero-shot setting is necessary here. What are the potential motivations behind this setting?

Minor comments:
1. Table 5: It would be better to denote T5-11B with 'zero-shot' mark but not only illustrate in line 201.


**After rebuttal**: I appreciate the authors addressed a few concerns, and thus I increase the score from 4 to 5.


**Clarity:**

There are several typos and grammatical issues. The authors are encouraged to revise their manuscripts accordingly.
1. line 74: public available -> publicly available
2. line 75: are showed -> are shown
3. line 77: resulting -> resulting in
4. line 80: assure that maximum number -> assure that the maximum number
5. line 100: please rephrase: 'we randomly assign of the settings'
6. Figure 1 (right): summury -> summary
7. line 290: an open-domain conversations -> open-domain conversations


**Documentation:**

Yes.

**Ethics:**

No.

**Relation To Prior Work:**

Yes.

**Summary And Contributions:**

This paper proposes a new dataset called QAConv for conversational question answering, which contains more than 34k QA pairs.
Different from previous works, QAConv contains relatively long conversations and various domains. The system should read the conversations and answer the relevant questions using a span or predict it as 'unanswerable', which is similar to the task setting in SQuAD 2.0. The dataset contains two tracks: chunk mode and full mode, which is determined by whether the conversational chunk is provided or not. The authors built various baseline systems on top of state-of-the-art pre-trained language models. The experimental results show that these baseline systems are about to reach human performance within a gap of 5% only.

---

> ### Author Response · Authors · 2021-07-08
> **Response**
>
> Thank you Reviewer AzsR for the feedback. We have modified the typo indicated in your Clarity section and we are happy to answer your questions and concerns.
>
> * Weakness (1) & Additional (4): Less challenging dataset?
> We would like to point out three reasons why we believe QAConv is a meaningful and challenging dataset.
>     * Our dataset has two modes, and the one you pointed out (79.99 v.s. 75.21) is under the chunk mode, which is assumed that the gold dialogue chunk is provided. However, the full mode is a more practical setting because in real use cases we will always face error propagation during the retrieval stage. Therefore, if you compare our full mode results in Table 8 (75.21 v.s. 51.44), we believe that there is still a big research gap to be mitigated.
>     * Large-scale pre-trained language models show promising performance on plenty of NLP tasks. The unified-QA model even got fine-tuned on 20 existing QA datasets based on T5 models. Comparing to many existing QA corpus that machines have “surpassed” humans, we believe that our chunk mode testing scenario can still provide some insights for pushing SOTA solutions for machine-reading comprehension on dialogues.
>     * Human performance numbers in the paper are more like zero-shot testing since we do not train those crowd workers to answer such questions and therefore it contains a certain variance of collected answers.
>
> * Weakness (2) & Additional (3, 5): Why zero-shot setting? & Why not report dev set?
> There are three reasons why we report zero-shot results. We will add these in a new paragraph in Section 3.3 after we finalize our discussion.
>      * By showing zero-shot performance, we would like to prove that there is a big gap between existing document-based QA tasks and the dialogue-based QA tasks we are proposing to solve. The knowledge is not that easy to be transferred like a plug-and-play setting.
>     * We show the gap between zero-shot and fine-tuning can also help us identify the usefulness of our collected data, suggesting that our data is meaningful. Without QAConv, all the existing SOTA models can have very limited ability to understand and predict dialogue QA pairs.
>    * Zero-shot (or few-shot) performance is a hot research topic in the whole research community. Even with training data available, it does not mean that we do not need to work in such a direction to improve optimization efficiency, data efficiency, and transferring ability. People simulate such hard scenarios very often in many tasks, for example, image classification tasks [1,2], intent detection [3,4], dialogue state tracking [5], dialogue response selection [6,7].
>
> About the development set, since we are not like some other datasets (e.g., SQuAD, GLUE, ShARC, etc) that hide their test set and only release the dev set, we originally did not include such development set numbers in the first draft. Thanks for letting us know the need and we have included those numbers in the Appendix (Table 12, 13, and 14).
>
> [1] A baseline for few-shot image classification
> [2] Rethinking few-shot image classification: a good embedding is all you need?
> [3] Discriminative nearest neighbor few-shot intent detection by transferring natural language inference
> [4] Cg-bert: Conditional text generation with bert for generalized few-shot intent detection
> [5] Transferable multi-domain state generator for task-oriented dialogue systems
> [6] TOD-BERT: pre-trained natural language understanding for task-oriented dialogue
> [7] Language models as few-shot learner for task-oriented dialogue systems

---

> > ### Author Response · Authors · 2021-07-08
> > **Response (Cont.)**
> >
> > * Addition (1): Multi-span Answers?
> >
> > The reason why our dataset can potentially contain multiple text span is that during the data collection period, the only constraint we have is forcing crowd workers can only write answers that are appeared in the conversations. Therefore, during analysis, we found out that there are around 0.5% to 1% of answers that are coming from multiple text spans.
> >
> > We include some examples in the Appendix (Table 16). For example, it happens when the answer contains pronouns, for example,  “our courts” → “the court” or “my hometown” → “John’s hometown” (we force crowd workers to not include first-person and second-person pronouns, line 104). Also, we have a question like “what can be awful but lawful?” and the gold answer is “officer involved shootings”, which has the original text like, “... David Klinger: There's a term of art called awful, but lawful. So sometimes officers are involved in shootings that don't really sound that good, but the law says it was an appropriate use of force, ...”
> >
> > For evaluation, since in the end, we are comparing two strings, the predicted one and the gold one, so there is no difference between single-span and multi-span answers.
> >
> > * Additional (2): Why BART for dialogue summarized?
> >
> > We use the BART model as our dialogue summarizer is because 1) we have the latest publication on dialogue summarization (https://arxiv.org/abs/2105.14064) and it is a BART-based model, and 2) the results achieved by the model is still the SOTA performance on SAMSum dataset.
> >
> > * Additional (Minor): T5-11B
> >
> > The results of T5-11B (UnifiedQA) are reported under the zero-shot column already. Can you elaborate more about what do you mean by “denote zero-shot mark”?

---

> > > ### Author Response · Authors · 2021-07-14
> > > **Final Check-in**
> > >
> > > Hi Reviewer AzsR,
> > >
> > > Just want to check in the post so you will not miss our responses. Did we answer your questions or concerns above? It will be great to hear from you, not only the rating but also your final suggestion.
> > >
> > > One last thing we would like to emphasize is that there is no such dataset as QAConv available in the field, which means that without our dataset, the best performance SOTA models can achieve is the "zero-shot" performance shown in Table 5 and Table 8. Hope our response can convince you that our dataset is important, meaningful, and influential to the NLP community, especially the dialogue + QA field.
> > >
> > > We are happy to provide more information if needed. Thank you.

---

### Official Review · Reviewer_Y8HH · 2021-07-02
**Novel dataset for QA over long conversations**

**Rating:** 7
**Confidence:** 4

**Strengths:**

Generally, the paper is well-presented and easily understood. The proposed dataset presents a clear contribution to the field, with the closest related datasets either focusing on shorter, chit-chat style dialogues, or containing multiple-choice answers instead of span-based/textual answers. The work highlights a shortcoming of common retriever models, which fail to perform well for retrieving information from long conversations; potentially, the release of this dataset could spur development in that direction. Additionally, the data collection approach of validating machine-generated questions rather than using human-generated questions may prove useful for further dataset construction.

**Weaknesses:**

The primary weakness of the paper may be the multi-hop questions. A drawback of the dataset construction strategy is that all questions seem to be generated over a single chunk of 512 tokens. While this is fine for single-hop questions, it means that all hops in multi-hop questions are through entities and sentences that appear very close together in the text. This dodges one of the primary challenges of multi-hop QA, retrieving very different documents about very different entities.

It is furthermore not clear to me to which degree the generated questions are actually multi-hop -- the authors use a question generation model trained on a multi-hop dataset, but it is conceivable (even likely) that such a model could generate single-hop questions. Given that, I would expect to see either a strong indication of the quality of the QG model, or an analysis of questions on this basis, e.g. how many questions are answerable through 1 hop, 2 hops, and so on (see also for example Table 3 of [1]).

These issues do not invalidate the primary contributions of the paper, but the claims of the dataset including multi-hop questions may be overstated. If the authors can convince me otherwise, I would be happy to adjust my score upwards.

EDIT: With the inclusion of a manual analysis of 50 sampled questions illustrating the degree to which the questions in this dataset are multi-hop questions, my concerns regarding overstatement have been addressed. I have adjusted my score accordingly.

[1] Yang et al., 2018. HotpotQA: A Dataset for Diverse, Explainable Multi-hop Question Answering.

**Additional Feedback:**

A few minor grammatical issues I noted:

line 267: "cloze-style QA that a model has to fill in the blanks" -> "cloze-style QA *where* a model has to fill in the blanks" (same error on lines 269 and 271)

line 280: "most close" -> "closest"

line 292: "recent proposed" -> "recently proposed"



**Clarity:**

The paper is generally well-written and easy to follow. A minor, useful improvement to clarity would be to specify whether the BART summarizer is abstractive or extractive.

**Correctness:**

The construction process and the empirical evaluation seem reasonable and appropriate.

**Documentation:**

The paper does not discuss maintenance of the dataset and leaderboard, nor is the license of the dataset itself mentioned (although the license can be seen on GitHub). Otherwise, sufficient documentation is included.

**Ethics:**

No immediate issues come to mind.

**Relation To Prior Work:**

The authors include a good discussion of prior work and how this dataset differentiates itself.

**Summary And Contributions:**

This paper introduces a new dataset of 34,204 question-answer pairs for question answering over long and information-rich conversations, a neglected topic in the literature. The task includes two modes, chunk and full, where systems attempt to answer questions based either on a chunk of the conversation or the full conversation. The paper furthermore proposes strong baselines in the form of RoBERTa- and T5-models. The gap between baselines and human performance in chunk mode is small; however, performance drops significantly in full mode, highlighting problems with common retrieval models. The dataset consists partially (41.2%) of machine-generated questions, which are subsequently modified by crowd workers; the efficacy of this approach is a secondary contribution of the paper.

---

> ### Author Response · Authors · 2021-07-08
> **Response**
>
> Thank you Reviewer Y8HH for the feedback. We have modified the typo indicated in your Additional Feedback section and we are happy to answer your questions and concerns.
>
> * Multi-hop questions?
>
> QAConv questions are limited to the context within the same chunk due to the length of long conversations. Unlike hotpotQA which explicitly merges context from different document sources using the Wikipedia hyperlink graph, we do not have such entity graph available. Thus, we use an implicit way, replying on multi-hop QG generators, to collect “potential” multi-hop questions. Without such a graph, it is also hard for us to measure the level of multi-hopping and come up with something like Table 3 in the HotpotQA paper. Moreover, since our machine-generated questions are just used as recommendations for crowd workers, we cannot guarantee that the level of reasoning in those final QG questions. The conclusion of the results in Table 3 is to show that QG models can successfully generate some “harder and better” questions. We add additional experiments in the Appendix (Table 17) to show that there is a remarkable performance gap between QG and Non-QG questions in the test set based on existing QA solutions.
>
> We’d like to argue that multi-hop questions do not necessarily need to come from different documents. As a length comparison, our input text has 303.5 words on average per chunk and hotpotQA has only 163.5 words on average (2 gold documents) after the retrieval stage. We manually check some questions and found out that our multi-hop QG model often generates questions that combine entities or conditions with the “AND” relationship. We provide some question examples we found that could be helpful for people to understand the potential level of reasoning in QAConv (Examples are added in the Appendix Table 18).
>
> * BART summarizer?
>
> The BART dialogue summarizer is an abstractive model, which is coming from our latest state-of-the-art results (https://arxiv.org/abs/2105.14064).
>
> * The Documentation?
>
> The maintenance of the dataset and leaderboard and the license of the dataset are all provided in the Appendix. If you click the “Supplementary Material” on the submission page, you should be able to get it.(https://openreview.net/attachment?id=QkOBP-aD1qA&name=supplementary_material).

---

> > ### Author Response · Authors · 2021-07-08
> > **Examples**
> >
> > Example 1
> > > Question: What person has the numbers for the Montana lawyers and is best qualified to explore the deal?
> >
> > > Partial Context (we skip some context in the chunk): Steve Duffy: ...,  willing to make to settle just the Montana case,\nbut it appears that their real interest would be in a \"global\" deal.  Any\ncomments?  SWD
> > Michael Burke: **Steve, Stan and I have discussed this and we agree that Mike Moran should\ntake the lead and explore all aspects of an Enron Global deal**.  I know that\nyou will assist Mike in this endeavor.  thanks, mike
> > Steve Duffy: Sounds good.  **Mike Moran has the numbers for our Montana lawyers and I will\nassist him any way I can**.  The big question is whether Enron, as a whole,\nwould be willing to give up any protection they might still have under the\nold  InterNorth policies.  SWD
> > …
> >
> > Example 2
> > > Question: "Who is the president of the country where Ofeibea quist-arcton is talking about the tensions and violence in Nigeria ?”
> >
> > > Partial Context (we skip some context in the chunk):
> > OFEIBEA QUIST-ARCTON, BYLINE: One woman we spoke to has lived here all her life. She was born here, married here, has children here. She said I'm going. I don't feel safe. You know, the ground was shaking when we heard those bombs. We don't feel ...
> > JENNIFER LUDDEN, HOST: **We are talking about the tensions and violence in Nigeria**. We'll have more with NPR's Ofeibea Quist-Arcton from Nigeria, and also former Ambassador John Campbell coming up. We'll also talk with an activist from Nigeria. If you have questions, ...
> > JENNIFER LUDDEN, HOST: This is TALK OF THE NATION from NPR News. I'm Jennifer Ludden. Nigeria has long faced challenges from corruption, an economy that relies on oil exports and simmering ethnic and religious tensions, tensions made evident in the recent series of bombings by Boko Haram, the militant ...
> > JENNIFER LUDDEN, HOST: **It's the latest crisis for President Goodluck Jonathan. We're talking today with Ofeibea Quist-Arcton, NPR's foreign correspondent, now in Kano, Nigeria**; and John Campbell, former U.S. ambassador and political counselor to Nigeria. He's now a senior fellow for Africa policy studies at the Council on Foreign Relations.
> >
> > Example 3
> > > Question: Which person was mentioned as the shooter in case A and B?
> >
> > > Partial Context (we skip some context in the chunk):
> > …
> > MR. FREEDMAN (RESPONDENT): … They both deserve the death penalty. They -- they were -- the prosecutors were aware that the -- the death penalty is what stirs the pot here, and so they were urging somebody to be the shooter to get the death penalty. If this wasn't a death penalty case, I don't think they -- it would have mattered who killed who. And so they were urging --
> > JUSTICE KENNEDY: Well, I think there's quite a difference in -- **in case A where you say our position is that Stumpf was the shooter**, pure and simple. That's it. **In case B, they say we think Stumpf was the shooter**. We're not 100 percent sure, but he should get the death penalty. The alternative is before the sentencer and the sentencer can make that determination.
> > …

---

### Official Review · Reviewer_pxGx · 2021-07-03

**Rating:** 6
**Confidence:** 4
**Correctness:** Correct and sound.
**Clarity:** very well written.

**Strengths:**

1. Novel dataset, which proposes a quite interesting new scenario.
2. Diverse collection pipeline to include both machine-generated and human-generated ones.
3. Comprehensive baselines for future research to compare against.

**Weaknesses:**

The ablation study is not quite enough. For example, how is the performance for separate human-generated and machine-generated questions. Is there any performance gap between them?

**Additional Feedback:**

Please provide more detailed ablation study about different annotation question types.

**Documentation:**

There are sufficient details provided.

**Ethics:**

no ethical concerns.

**Relation To Prior Work:**

clearly discussed.

**Summary And Contributions:**

This paper introduces a new dataset, which is based on a conversation between multiple users as the knowledge source to conduct question answering task. This dataset differentiates from the previous ConvQA dataset in the sense that it uses longer, realistic, and complex real-life dialogs like emails, discussion forum, etc.  Thus, the dataset has a great contribution to the community. The paper conducts very comprehensive experiments using different pre-trained models. The results are quite promising for both chunk mode and full mode. I believe the new dataset could serve as a valuable asset for the QA  and dialog community to solve and will bring real values to industries who are enthusiastic about understanding user dialog records, etc.

(Rating has been modified)

---

> ### Author Response · Authors · 2021-07-08
> **Response**
>
> Thank you Reviewer pxGx for the feedback. We are happy to answer your questions and concerns about the ablation study.
>
> * Is there a performance gap between HW and MG questions?
>
> We found out that QG questions, in general, perform worse than HW questions. This further confirms our hypothesis and the results are shown below and in the Appendix (Table 17) as well.
>
> TEST SET
>
> QG
>
> | exact  |   f1 |   fzr |   total |   Name |
> | -- | -- | -- | -- | --|
> | 42.94 | 47.67 | 61.80   |  1232 |   distilbert-base-uncased-zero-shot |
> | 52.44|  57.77 | 69.28  |   1232  |  roberta-base-zero-shot|
> |52.60 | 58.57 | 69.69   |  1232  |  roberta-large-zero-shot|
> |34.90 | 40.17 | 56.51   |  1232  |  bert-base-cased-zero-shot|
> | 54.79 | 60.46 | 70.99   |  1232  |  bert-large-uncased-whole-word-masking-zero-shot|
> | 64.04 | 72.02 | 78.79   |  1232  |  distilbert-finetuned|
> |65.18 | 73.50 | 79.57   |  1232  |  bert-base-finetuned|
> | 65.02 | 73.60 | 80.26   |  1232  |  bert-large-finetuned|
> | 70.05 | 77.93 | 83.14   |  1232  |  roberta-base-finetuned|
> | 68.02 | 75.31 | 81.33   |  1232  |  roberta-large-finetuned|
> | 73.05 | 80.97 | 85.69   |  1232  |  roberta-large-largebsz-finetuned|
> | 71.59 | 78.58 | 83.78   |  1232  |  bert-large-largebsz-finetuned|
>
> Non QG
>
>   | exact  |   f1 |   fzr |   total |   Name
> | -- | -- | -- | -- | --|
>  | 48.44 | 55.58 | 66.26 |    2271  |  distilbert-base-uncased-zero-shot|
>   |60.63 | 68.20 | 75.70 |    2271  |  roberta-base-zero-shot|
>  | 62.53 | 70.86 | 77.47 |    2271  |  roberta-large-zero-shot|
>  | 46.98 | 54.81 | 65.67 |    2271  |  bert-base-cased-zero-shot|
>  | 64.46 | 72.26 | 78.51 |    2271  |  bert-large-uncased-whole-word-masking-zero-shot|
>  | 63.50 | 74.99 | 80.40 |    2271  |  distilbert-finetuned|
>  | 67.02 | 77.81 | 82.94  |   2271  |  bert-base-finetuned|
>  | 64.99 | 76.64 | 81.97  |   2271  |  bert-large-finetuned|
>  | 71.73 | 81.67 | 85.90  |   2271  |  roberta-base-finetuned|
>  | 68.60 | 79.15 | 84.15  |   2271  |  roberta-large-finetuned|
>  | 75.47 | 85.11 | 88.74  |   2271   | roberta-large-largebsz-finetuned|
>
>
> * Ablation study of different question types
>
> We showed that 35.5% of errors are what-question, 18.2% are which-question errors, 12.1% are how-question errors, and 10.3% are who-question errors (Section 4). These numbers are obtained using RoBERTa-Large after fine-tuning on our QAConv training set. Please feel free to let us know if you would like to know more about the error analysis.

---

> > ### Comment · Reviewer_pxGx · 2021-07-12
> > **Response**
> >
> > Thank you for posting the response. From the results, I can see the machine-generated questions are actually harder than human-written ones. Can you explain why this is happening? I felt that human-written ones are more diverse, why is it even easier than machine-generated ones.
> >
> > Also, I have seen the other reviewers' concerns.
> > One concern I found from the reviewer Y8HH very valid is "A drawback of the dataset construction strategy is that all questions seem to be generated over a single chunk of 512 tokens. While this is fine for single-hop questions, it means that all hops in multi-hop questions are through entities and sentences that appear very close together in the text."  It seems that the datasets' questions are performing very limited multi-hop reasoning.
> >
> > Another concern I found from the reviewer AzsR very valid is "I am quite disappointed that current state-of-the-art models only have a minor gap to human performance. This was also indicated by the authors in line 224-226." Based on the results shown in the paper, the gap is too small, while the room for future improvement is quite limited.
> >
> > Based on the mentioned concerns, I decided to lower my score to 6.

---

> > > ### Author Response · Authors · 2021-07-12
> > > **Regarding your concerns and the new score**
> > >
> > > Hi Reviewer pxGx,
> > >
> > > First, we have a question generator trained on the multi-hop QA dataset because we would like to guide and recommend humans to write/modify and get harder questions. Human written questions are hard to guarantee the hardness of the questions, and most of the crowd workers will take an entity (e.g., Orlando) as an answer and then write a straightforward question (e.g., where did the IEFT meeting take place?). Thus, it is not surprising that crowd workers can just choose or modify our QG recommended questions and get harder ones we expected.
> > >
> > > Second, about the source of the question, as we replied to Reviewer Y8HH, one conversation chunk we have 303.5 words on average, and the hotpotQA has only 163.5 words on average (2 gold documents) after the retrieval stage. Thus, we agree that the long length of conversations (i.e., max 19,917 words) limits the way we collect "fully contextualized" QA pairs, but we do not agree that within each chunk we cannot provide challenging and meaningful (multi-hop) questions. Please check Table 18 in the Appendix for such examples.
> > >
> > > Third, as we replied to Reviewer AzsR, please check the three reasons below why we believe QAConv is a meaningful and challenging dataset:
> > >
> > > * Our dataset has two modes, and the one Reviewer AzsR pointed out (79.99 v.s. 75.21) is under the chunk mode, which is assumed that the gold dialogue chunk is provided. However, the full mode is a more practical setting because in real use cases we will always face error propagation during the retrieval stage. Therefore, if you compare our full mode results in Table 8 (79.99 v.s. 51.44), we believe that there is still a big research gap to be mitigated.
> > >
> > > * Large-scale pre-trained language models show promising performance on plenty of NLP tasks. The unified-QA model even got fine-tuned on 20 existing QA datasets based on T5 models. Compared to many existing QA corpus that machines have “surpassed” humans, we believe that our chunk mode testing scenario can still provide insights for pushing SOTA solutions for machine-reading comprehension on dialogues.
> > >
> > > * Human performance numbers in the paper are more like zero-shot testing since we do not train those crowd workers to answer such questions and therefore it contains a certain variance of collected answers.
> > >
> > > There is no dataset like this currently available in the field, which means that without our dataset, the best performance SOTA models can achieve is the "zero-shot" performance shown in Table 5 and Table 8. Hope our response can convince you that our dataset is important, meaningful, and influential to the NLP community, especially the dialogue + QA field.
> > >
> > > Happy to take any questions if you have any. Thank you.

---

> > > > ### Author Response · Authors · 2021-07-14
> > > > **Final Check-in**
> > > >
> > > > Hi Reviewer pxGx,
> > > >
> > > > Just want to check in the post so you will not miss our responses and our discussion with other reviewers as well. Did we answer your questions or concerns above, regarding multi-hopping and the challenging parts? It will be great to hear from you, not only the rating but also your final suggestion.
> > > >
> > > > Regarding multi-hopping concerns, as we discussed with Reviewer Y8HH below, he/she believes the level of reasoning is clear by checking the examples in Table 18. Also, we manually checked 50 machine-recommended QA pairs and provided the ratio of multi-hop types (https://openreview.net/forum?id=QkOBP-aD1qA&noteId=B6Xuj9SmHmg), showing that 40% of such examples are multi-hopping. As a side note, we would like to defend ourselves that we are not trying to propose a “multi-hop dialogue+QA dataset” but we try our best to increase the potential of our dialogue+QA data to include and cover such challenging and multi-hop cases. We will make this clear in the final paper.
> > > >
> > > > We are happy to provide more information if needed. Thank you.

---

### Comment · Area_Chair_npL7 · 2021-07-21
**Gap between development / test performance**

Hi Authors!

Thanks for providing development set results. We observe a large performance gap between the development set/test set (performance worse on the development set by roughly 10 EM). The appendix said it was a random split, but the performance gap makes me wonder otherwise. Also, is there a reason not to host a hidden test set?

---

### Decision · Program_Chairs · 2021-07-26

**Decision:**

Reject

**Comment:**

This paper introduces large-scale data studying question answering based on conversation history. The questions are collected either by automatically generated and then edited, or human written. The dataset poses a new benchmark to study dialogue understanding, both for training and benchmarking purposes.

The experimental results suggest that the main challenges of the benchmark come from paragraph retrieval, which can be studied in existing QA datasets such as NQ. As reviewers pointed out, the gap between human performance and model performance is minimal in the chunk setting, where the evidence paragraph is identified. Since multiple reviewers brought the size of the gap between chunk setting performance vs. full setting, the authors might want to de-emphasize chunk setting and present full setting as the main task.

While the authors pose "multi-span" as a unique contribution, questions containing multi-span annotations are very rare (1%). I would suggest pruning this, as the dataset does not really support studying multi-span answers as it occurs so rarely. The dataset construction largely follows recent QA dataset constructions, and there's little novelty except for the different types of evidence documents (conversation vs. Wikipedia document). If authors can add additional analysis on new challenges posed by using dialogue as an evidence document (vs. wikipedia text), the paper will be stronger. Also, the result breakdown by different types of dialogue corpus would be interesting.

The evaluation set up (gap between dev/test set) (thanks authors for answering my question on the supplementary material) should be clearly explained in main text. I am not persuaded why they did not clean the development dataset the same way they treated test data, as both are evaluation data.

The paper has merits, but would benefit from another round of re-writing and analysis.